# Alternative Non-Mammalian Animal and Cellular Methods for the Study of Host–Fungal Interactions

**DOI:** 10.3390/jof9090943

**Published:** 2023-09-19

**Authors:** Ana Marisa Fusco-Almeida, Samanta de Matos Silva, Kelvin Sousa dos Santos, Marcos William de Lima Gualque, Carolina Orlando Vaso, Angélica Romão Carvalho, Kaila Petrolina Medina-Alarcón, Ana Carolina Moreira da Silva Pires, Jenyffie Araújo Belizario, Lígia de Souza Fernandes, Andrei Moroz, Luis R. Martinez, Orville Hernandez Ruiz, Ángel González, Maria José Soares Mendes-Giannini

**Affiliations:** 1Department of Clinical Analysis, School of Pharmaceutical Science, Universidade Estadual Paulista (UNESP), Araraquara 14800-903, SP, Brazil; ana.marisa@unesp.br (A.M.F.-A.); samanta.matos@unesp.br (S.d.M.S.); k.santos@unesp.br (K.S.d.S.); mw.gualque@unesp.br (M.W.d.L.G.); c.vaso@unesp.br (C.O.V.); romao.carvalho@unesp.br (A.R.C.); kaila.medina@unesp.br (K.P.M.-A.); carolina.pires@unesp.br (A.C.M.d.S.P.); jenyffie.belizario@unesp.br (J.A.B.); ligia.fernandes@unesp.br (L.d.S.F.); andrei.moroz@unesp.br (A.M.); 2Basic and Applied Microbiology Group (MICROBA), School of Microbiology, Universidad de Antioquia, Medellin 050010, Colombia; orville.hernandez@udea.edu.co (O.H.R.); angel.gonzalez@udea.edu.co (Á.G.); 3Department of Oral Biology, College of Dentistry, University of Florida, Gainesville, FL 32610, USA; lmartinez@dental.ufl.edu; 4Emerging Pathogens Institute, University of Florida, Gainesville, FL 32610, USA; 5Center for Immunology and Transplantation, University of Florida, Gainesville, FL 32610, USA; 6Center for Translational Research in Neurodegenerative Disease, University of Florida, Gainesville, FL 32610, USA; 7Cellular and Molecular Biology Group University of Antioquia, Corporation for Biological Research, Medellin 050010, Colombia

**Keywords:** alternative methods, three-dimensional culture, *Danio rerio*, *C. elegans*, *Galleria mellonella*, *T. molitor*

## Abstract

In the study of fungal pathogenesis, alternative methods have gained prominence due to recent global legislation restricting the use of mammalian animals in research. The principle of the 3 Rs (replacement, reduction, and refinement) is integrated into regulations and guidelines governing animal experimentation in nearly all countries. This principle advocates substituting vertebrate animals with other invertebrate organisms, embryos, microorganisms, or cell cultures. This review addresses host–fungus interactions by employing three-dimensional (3D) cultures, which offer more faithful replication of the in vivo environment, and by utilizing alternative animal models to replace traditional mammals. Among these alternative models, species like *Caenorhabditis elegans* and *Danio rerio* share approximately 75% of their genes with humans. Furthermore, models such as *Galleria mellonella* and *Tenebrio molitor* demonstrate similarities in their innate immune systems as well as anatomical and physiological barriers, resembling those found in mammalian organisms.

## 1. Introduction

In recent years, the incidence of fungal diseases has significantly increased due to a rise in the number of individuals with severely compromised immune systems, such as those with HIV/AIDS, undergoing transplants, having a high incidence of cancer, asthma, tuberculosis, chronic obstructive pulmonary disease (COPD), and more recently, COVID-19 infections [1,2,3,4]. Systemic and endemic fungal infections caused by *Candida*, *Cryptococcus*, *Histoplasma*, *Blastomyces*, *Paracoccidioides*, *Coccidioides*, and other opportunistic diseases acquired from molds, such as aspergillosis, fusariosis, and mucormycosis, have shown an upward trend in recent years. For instance, aspergillosis alone affects over 24 million people annually [1]. Infections caused by dermatophytes and *Candida* spp. persist globally [5,6]. Epidemiological studies reveal that four million people suffer from oral, esophageal, and invasive candidiasis yearly, with almost 134 million affected by vulvovaginal candidiasis [1]. Fungal diseases are associated with high mortality rates, and instances of resistant strains are on the rise [7]. Furthermore, research focusing on the pathophysiology of fungal infections needs further exploration to identify new therapeutic targets and develop novel classes of antifungal drugs [8,9].

Simultaneously, the quest for new non-mammalian animals and alternative cellular methods to study mycoses aligns with scientific progress in understanding fungal diseases and, concurrently, emphasizes the well-being of vertebrate animals. Research animals are indispensable for advancing knowledge across diverse domains, from elucidating behaviors to comprehending diseases, disorders, drug development, and other pursuits [10]. However, many countries have enacted legislation to curtail mammalian usage in specific sectors, including animal experimentation, thereby intensifying the scrutiny of these methodologies. The 3 Rs principle (replacement, reduction, and refinement) has been incorporated into legislation and animal experimentation guidelines across numerous countries [8,11]. These principles advocate replacing research animals with systems and strategies that yield more robust and dependable data [8].

In each research field, strategies must be devised to harmonize reduced time and cost with alternative approaches to mammalian utilization [12]. Alternative non-mammalian animals such as three-dimensional cultures, *D. rerio*, *C. elegans*, *G. mellonella*, and *T. molitor* have been embraced in fungal pathogenesis research to supplant vertebrate usage (Figure 1).

The 3 Rs endeavor to embrace alternatives for animal safeguarding, minimize animal numbers without undermining experiment reliability, and ameliorate animal welfare and health during scientific investigations [10,13]—refinement advocates for enhancing animal welfare and health during scientific studies.

Concerning the first “R” (replacement), the recommendation is to employ alternative organisms such as invertebrates, microorganisms, and in silico methods as substitutes for vertebrates, vertebrate embryos, and cells of vertebrate origin. Tissue engineering can reconstruct tissues/organs with fundamental characteristics through cell cultures. In silico computational methods facilitate the construction of computational models that aid disease comprehension. Moreover, alternative animals like *D. rerio*, *C. elegans*, *G. mellonella*, and *T. molitor*, which possess less complex nervous systems, have also found utility [10,11,13].

The second “R” pertains to reducing the number of animals used in experiments. However, this reduction should not compromise experiment quality. Complementary techniques, such as cell cultures, utilization of other organisms, and computer models, should always be amalgamated. Lastly, the third “R” concerns refinement, which underscores enhancing animal treatment to provide well-being essential for healthy biological functioning and typical behavioral patterns. Enhancing procedures to mitigate encroachment or stress levels and ameliorating housing and animal husbandry should be the objective of every scientist [10,14,15].

This review delves into alternative models, including three-dimensional (3D) cultures, *D. rerio*, *C. elegans*, *G. mellonella*, and *T. molitor*, which have been employed in various studies to assess mechanisms of host–fungal interactions.

## 2. Cell Culture

In recent decades, cell culture has emerged as one of the most extensively employed techniques for substituting, preceding, or complementing the use of animals in research. In the 20th century, cell culture was developed to study animal cell behavior outside the body in a controlled environment. Initially, it employed crude extracts and animal lymph as culture mediums. However, it has since evolved into an indispensable tool for investigating degenerative diseases, cell therapy, stem cell research, testing the toxicity and efficacy of novel compounds, and understanding pathogenesis, among other applications [16,17].

The predominant cell culture model employed in examining fungal pathogenesis is the monolayer (2D) culture due to its convenient alteration of culture mediums, subculturing, analysis, and standardization. Through phagocytosis assays involving macrophages [18], multiple researchers have discovered that phagocytic cells are more inclined to engulf planktonic cells as opposed to cells originating from biofilms, such as *Cryptococcus neoformans* and *Candida albicans* [18,19]. Some investigations have centered around elucidating the behavior of exocytosed macromolecules from fungi during cellular infection. For instance, exoantigens extracted from the fungus *P. brasiliensis* stimulated the proliferation of pulmonary fibroblasts in vitro. In contrast, a mannoprotein extracted from the cell wall of *C. albicans* induced alterations in the cell cycle and inhibited apoptosis in human keratinocyte cells (HaCat) [20]. Cell cultures also offer insights into interaction studies. For example, some researchers have demonstrated the relevance of the Gmp1 binding protein in immune evasion, cell adhesion, invasion, and colonization of *C. albicans* in human umbilical vein endothelial (HUVEC) and HaCat host cells. The utilization of primary or lineage-specific cells from human and mammalian hosts (macrophages, neutrophils, dendritic, endothelial, epithelial, and stem cells, among others) has facilitated comprehension of the fungus–host relationship and the pursuit of prospective therapeutic targets [21,22,23].

Nevertheless, this kind of culture model has limitations due to its simplistic architecture, leading to diminished physiological functionality caused by the decline of characteristics during culturing. Despite its close resemblance to natural conditions, the in vitro process still presents challenges for cell development. Its cell–cell and cell–matrix adhesion is diminished, and it lacks the attributes (such as heterogeneity and three-dimensional architecture) of an in vivo tissue since its nutritional and hormonal environment undergo modification [17,24].

Thus, three-dimensional (3D) cell cultures are extensively employed for novel drug screening of antifungals. Several advantages are tied to obtaining more precise toxicity and efficacy results than in vivo models, along with host–pathogen interactions closely mimicking reality [25,26]. Recently, pulmonary spheroids have been employed to screen broad-spectrum antifungal drugs, yielding outcomes that indicate three-dimensional cell cultures are less susceptible to the toxic effects of drug treatment compared to monolayer cultures. Furthermore, it was observed that monolayer cell cultures are approximately ten times more sensitive than their 3D counterparts, underscoring that 3D cell cultures better resemble in vivo models. Consequently, three-dimensional (3D) cultures have emerged as a viable alternative for replicating the in vivo environment [26].

### 2.1. Three-Dimensional Cultures

Three-dimensional (3D) cultures are pivotal in vitro studies as an alternative model that curtails research expenses [24,27]. Although exploration of 3D culture began around the 1970s, historical records indicate that early 20th-century approaches to cell differentiation and aggregate models were already underway [27,28,29]. Nonetheless, during the late 20th and early 21st centuries, 3D cultures gained prominence, notably following an article by Weaver et al. in 1997 [30], followed by subsequent publications that underscored the significance of 3D culture, particularly in advancing tumor research [30,31,32].

Numerous researchers demonstrated that specific antibodies could hinder a surface receptor for β1-integrin, causing cancer cells to mimic normal cells, a phenomenon not observed in a 2D environment. This event marked the continued attention on 3D cultures, initially for tumor studies and subsequently in fields such as toxicology, pathogenesis, disease physiology, and other applications [33,34].

Evidence indicates that 3D cultures can create a microenvironment more akin to that of animals than 2D cultures: cells in 3D settings exhibit distinct morphology and physiology from their 2D counterparts (Figure 2). Moreover, cellular responses, proliferation rates, gene and protein expression levels, and susceptibility to molecules render 3D cultures more akin to responses obtained within an in vivo environment, differing from those witnessed in 2D models [24,33,35,36].

Diverse methodologies exist for creating 3D cultures, although not all techniques have been validated for studying fungal pathogenesis. The principal methodologies include scaffold-free spheroids; spheroids formed using scaffolds, organoids, and skin/mucosa models. The scaffold-free spheroid approach requires no material to form 3D cultures; cells autonomously organize and naturally connect [38,39,40]. The established methodologies include suspended droplet microplates, magnetic levitation, and spheroid microplates with ultra-low fixation coating [41]. This approach is straightforward, demanding less economic investment and time than animal experiments and obviating the need for ethical committee approval. Consequently, spheroids facilitate cell interaction, aggregation, and the establishment of 3D cultures [42,43].

Conversely, spheroids formed through scaffolds necessitate hydrogels, polymeric materials characterized by varying physical, chemical, and biological attributes [44,45]. Prominent hydrogels in 3D cultures include alginate, collagen, agarose, and natural polymers. However, some scaffolds can be fashioned from synthetic hydrogels. These hydrogels enclose cells in the culture medium, forming miniature pearls through their pores. This arrangement permits the entry of nutrients and waste products exit alongside favorable biocompatibility with the culture medium [33,34,44].

Organoids entail multilayered cell structures enveloped by an extracellular matrix that safeguards cellular connections and delineates tissue functions and structures. Organoids are self-arranging cells that closely mimic human organs (Figure 3). For sustained organoid function and differentiation, a culture medium is necessary. Two types of stem cells—embryonic pluripotent stem cells (ESCs), governing embryonic organ development, and adult somatic stem cells (ASCs), specific to maintaining homeostasis and regenerating mature organs—give rise to organoids. The culture platform can reside on a biopolymer surface where cells are distributed in culture flasks coated with extracellular matrix molecules like collagen I and Matrigel. Alternatively, cells are encapsulated within a natural or synthetic biopolymer gel to facilitate more physiological 3D organization. Culture on mechanical supports, organs, or organ fragments is maintained at the liquid–air interface to facilitate gas exchange, and cells are grown on a porous membrane at this interface [46,47].

Skin 3D models, also called artificial skin or reconstructed skin, replicate the layers constituting human skin—a complex structure encompassing three layers: the epidermis, dermis, and hypodermis [47,48,49]. The epidermis predominantly comprises keratinocytes, Langerhans cells, and melanocytes that contribute to microbial defense, melanin synthesis for radiation protection, and endocrine functions. The dermis houses fibroblasts, synthesizing the extracellular matrix, in addition to endothelial cells, macrophages, and various components such as blood vessels, sebaceous glands, sweat glands, hair follicles, and nerves, collectively upholding physiological activities. The hypodermis, rich in adipocytes, offers mechanical protection and energy storage alongside playing a pivotal role in temperature regulation [49,50,51].

Skin 3D cultures effectively recreate the epidermal and dermal components via cells and biopolymers for skin reconstruction. A frequently utilized methodology involves employing biopolymers to construct skin layers, enabling the incorporation of fibroblasts, keratinocytes, stem/progenitor cells, or reprogrammed cells, thereby mimicking the tissue’s morphological and biological characteristics. Alternatively, 3D printers can assemble the cellular layers [47,49,50,52]. In recent years, the burgeoning demand for skin models has propelled the rapid expansion of the market, resulting in commercially available skin models such as EpiDermTM (MatTek, Ashland, MA, USA) and EpiskinTM (formerly by Episkin, Chaponost, France; now by L’Oreal, SkinEthic, Nice, France), among others [12,53,54,55,56].

Aside from skin models, 3D mucosal cultures have also been reported, particularly in models concerning oral mucosa and other mucous membranes encompassing the organism [57]. The oral mucosa comprises stratified epithelium that safeguards against physical and chemical harm, microorganisms, and dehydration. It also encompasses a basement membrane composed of thin layers of extracellular matrix, type IV collagen, and laminin, contributing to protection and homeostasis [58,59,60]. Analogous to skin models, these cultures exhibit organizational complexity, effectively simulating an in vivo environment [57].

3D mucosal cultures, similar to skin models, can also be created through scaffolds to generate cellular layers or procured commercially. At least two available models include one featuring oral squamous cell carcinoma (Skinethic Laboratory, Nice, France) and another comprising primary oral keratinocytes (MaTek Corporation, Ashland, MA, USA). These models are cultured in chemically defined mediums in an air–liquid interface configuration to mimic oral epithelial tissue, with or without keratinization [57]. Each 3D cultivation method presents advantages and disadvantages, contingent on the desired experimental outcomes. The employment of 3D cultures fosters a highly physiological environment characterized by direct cell–cell contact and interactions with extracellular matrix components, including collagen I, collagen III, and fibronectin, which are principal constituents [16,35].

### 2.2. Three-Dimensional Cultures in Fungal Pathogenesis

#### 2.2.1. Organoids

One of the primary attributes of organoid models is their capacity to replicate pathogen–host interactions, with current studies primarily centered on virology, particularly in the context of brain organoid models and the Zika virus [61,62,63]. Moreover, recent research has explored the application of organoid brain technology to develop drugs for preventing or treating Zika infections [64,65,66,67]. Organoids derived from intestinal and gastrointestinal tissues have also found utility in investigating pathogen–host interactions for various pathogens, including rotavirus, enteric bacteria, *Clostridium difficile*, *Helicobacter pylori* [68,69], and norovirus [70,71,72].

However, a paucity of literature addresses the utilization of organoid models for studying fungal–host interactions. Demonstrations have been made that the pulmonary and intestinal organoid model exhibited cytokine release evaluation after infection with heat-inactivated *A. fumigatus* and other strains. Moreover, a notable rise in the expression of biologically significant Toll-like receptors elicited a range of pro-inflammatory mediators, signifying the model’s efficacy in scrutinizing and quantifying immune responses [73].

Employing organoids derived from esophageal and gastric cancer cells to explore the interaction of fungal plaques revealed that the microenvironment favored the growth and invasive fungal infection by *Candida* spp. [73]. The use of explant culture from murine intestinal mucosa exhibited a highly prismatic epithelium with specific cell-to-cell epithelial connections, a basal lamina, and diverse types of connective tissue cells. This approach evaluated the infection process with *C. albicans*, revealing its capacity to adhere, penetrate, and infiltrate the epithelial barrier [74].

Lung organoids have been harnessed to probe *Pneumocystis* spp. infections; organisms predominantly confined to the alveolar lumen. This study extracted type 1 and 2 lung epithelial cells through lung digestion, followed by organoid development. The insights garnered facilitated establishing a mean organoid size standard for introducing target microorganisms; smaller organoids tended to collapse during the infection [75].

However, akin to any model, organoids possess limitations. Notably, they lack innervation, blood vessels, and immune system cells, and the disease processes they replicate might not fully parallel those observed in the human body [76]. The self-organizing characteristics of organoids present the prospect of incorporating additional cellular or microbial elements, and although their exploration has increased due to their human-organ-like resemblance, there are still gaps in understanding many approaches. Particularly in studies employing organoids as models for fungal infections, substantial avenues remain unexplored [75,76,77].

#### 2.2.2. Skin and Mucous Membrane Model

Comprehending the conduct and adaptation of fungal pathogens within the human body has surged in importance, given the escalating prevalence of fungal infections. Recent investigations have led to the development of 3D models representing epithelial tissues designed to scrutinize fungal infections of the respiratory and oral mucosal regions.

A 3D epithelial model using a transwell scaffold was deployed to investigate upper and lower respiratory tract infections. Infection with *A. fumigatus* within this model revealed the detection of pathogens, phagocytosis, and migratory characteristics, illustrating the functionality of this in vitro system in simulating the in vivo environment [78].

Creating a reconstituted oral mucosa model utilizing fibroblasts, type IV collagen, and keratinocytes extracted from gingival biopsies facilitated infections with *C. albicans* and *S. aureus*, simulating co-infections within the oral mucosa. The model exhibited surface damage upon *C. albicans* infection alone and invasion of the subepithelial collagen matrix in the case of co-infection [60].

This reconstituted oral mucosa model further highlighted fungal infection behavior, both with and without the addition of polymorphonuclear cells. Including immune cells exhibited protective effects and reduced epithelial damage, implying a Th1-linked immune response [79].

A recent study featuring a 3D co-culture model of oral mucosa infected with *Candida* spp. investigated the efficacy of silver microcrystals. This model demonstrated satisfactory biocompatibility, and the tested compound exhibited promising effectiveness in treating fungal infections [79].

These investigations are pivotal for comprehending the host’s defense mechanisms against pathogens. Nonetheless, the validation of 3D models as reliable predictors of biological responses is necessary, and a demand persists for more robust models capable of mimicking the in vivo environment while diminishing the need for mammalian usage in research, aligning with the principles of the 3Rs.

## 3. The *Danio rerio* Model

*Danio rerio*, commonly known as zebrafish, belongs to the Cyprinidae family, including other fish species like carp, goldfish, barbs, and related genera [80,81]. The initial documentation of this fish dates back to 1822 when Francis Hamilton included it in his manuscript “An Account of the Fishes Found in the River Ganges and Its Branches” [82]. Indigenous to rivers and streams in India, Nepal, and Bangladesh, this small fish thrives in vegetated lakes, rice paddies, irrigation canals, and streams with gentle currents [83]. *D. rerio* prefers forming groups in its natural habitat, a behavior also observed in captivity [84]. In laboratory settings, the typical temperature is around 28 °C, but wild strains endure temperatures ranging from 24.6 °C to 38.6 °C without experiencing heat stress [81]. Its diet primarily consists of insects, nematodes, and other zooplankton [85].

In the wild, *D. rerio* mainly reproduces between April and August. However, in captivity, their reproduction can occur year round [84]. Light significantly influences the reproductive behavior of *D. rerio*. While they tend to initiate their reproductive activities at dawn, their circadian rhythm may vary in laboratory environments. For successful fertilization, males and females coordinate the release of sperm and egg-laying. A mature female can release hundreds to over a thousand oocytes [86]. Parents often consume eggs in nature, but in laboratories, measures like mesh barriers, notched barriers, or marbles prevent egg consumption by parents. Additionally, individual or group breeders are commercially available, where selected fish are utilized for reproduction and returned to their respective aquariums [81].

After external fertilization, the progeny are considered embryos until they hatch or are born. The embryonic stage lasts up to 72 h, during which the fish’s mouth transitions to the protrusion stage [87]. The embryo undergoes synchronous cell cycles during development, with each cell division taking around 15 min. These divisions result in cell differentiation, giving rise to the embryo’s various systems and tissues. The fish’s swim bladder inflates at approximately three to four days, enabling vertical swimming and feeding. The next four weeks are spent in the larval stage [87,88]. Variables such as temperature, density, and individual differences can complicate the standardization of larval stages. Upon reaching a size of approximately 11 mm after four weeks, the fish are classified as juveniles and usually achieve sexual maturity between 10 and 12 weeks post-fertilization. Their lifespan depends on breeding conditions, with an average of 3.5 to 5 years. Signs of aging can be observed from the second year of life [81].

Over the past three decades, *D. rerio* has emerged as a pivotal vertebrate model in the scientific literature, contributing to studies in genetics, development, physiology, behavior, and evolution [87,89]. This fish’s genetic utility, combined with its rapid extrauterine development, prolific proliferation, and transparent embryos, has established *D. rerio* as an indispensable model for scientific research. Additionally, their hardiness, ease of maintenance, and cost-effectiveness further enhance their appeal [80,89].

*D. rerio*’s advantages in genetic studies have been recognized [86]. Pioneering genetic studies with *D. rerio* involved using gamma rays to randomly mutate the fish’s DNA, resulting in offspring with unique phenotypes, such as pigmentation defects. Over years of genome exploration by researchers, mutations in vertebrate sperm led to the identification of 4000 mutant strains exhibiting alterations in various aspects of development or embryonic behavior. This genetic insight significantly advanced vertebrate genomic discovery. The fish genome comprises around 26,000 genes within 25 pairs of chromosomes, encompassing roughly the same number of protein-coding genes as the human genome in about half the size but with a similar number of chromosomes [86].

The complete sequencing of *D. rerio*’s genome and its homologous genes to humans has sparked interest in identifying mutations and crucial functional genes [81,89]. With a well-defined genome containing 71% of genes orthologous to human proteins (or 82% considering disease-related proteins), *D. rerio* presents a powerful resource. Orthologs between fish and human counterparts are notably prevalent within functional domains. For instance, glucocorticoid receptors in *D. rerio* share 54% sequence similarity with their human counterparts, while thyroid receptors exhibit 91% sequence similarity [90,91].

The benefits of utilizing intact vertebrates for screening purposes are evident. Fish larvae possess functional livers, kidneys, and blood-brain barriers. These attributes contribute to the study of numerous compounds identified through *D. rerio* assays, with a seamless translation into mammalian in vivo models requiring minimal pharmacological optimization [91].

Experiments with this model can encompass embryos, larvae, or adults, spanning a diverse range of biological processes with fully integrated vertebrate systems and organs. In this context, therapeutic targets, physiology, drug metabolism, and pharmacology can be studied from the earliest stages of life. Moreover, the model allows for investigations into pain, sedation, tumor metastasis, vascular tone, intestinal motility, and more [89,91]. Many drugs used to treat human ailments exhibit analogous effects in *D. rerio* [90]. For example, drugs modulating glucose homeostasis in humans can be tested in fish, given its pancreas contains islets comprising alpha, beta, omega, and epsilon cells responsible for regulating glucose homeostasis by releasing glucagon, insulin, somatostatin, and ghrelin [91].

While wild *D. rerio* offers remarkable versatility, mutant, transgenic, and morphing models open an entirely new realm of possibilities in research. The evolution of gene-editing techniques, like CRISPR and morpholinos, has enabled the generation of organisms expressing fluorescent proteins, allowing for gene manipulation with knockdown, knockout, or overexpression to mimic specific diseases and conditions [92].

### D. rerio as a Model of Fungal Infections

Studies aiming to unravel the characteristics and behavior of fungal infections have been extensively documented in the scientific literature, utilizing different life stages of the zebrafish (*Danio rerio*) and capitalizing on its general advantages as a vertebrate model and its specific attributes, such as the transparency of young embryo larvae [93]. Among the fungi extensively investigated in the scientific literature, *C. albicans* is one of the most prominent subjects. This commensal fungus holds significant public health importance because it can cause various infections under specific conditions, particularly in immunocompromised host organisms [94]. Candidiasis can manifest as infections that range from mild and localized to severe cases involving dissemination to various organs and systems, which poses a substantial concern, particularly in nosocomial environments [94,95,96].

Transgenic *D. rerio* larvae have been employed as a model to uncover two strategies of *C. albicans* dissemination. The first strategy, coined “Trojan Horse” propagation, relies on phagocytes. In this mode, yeasts are phagocytosed by immune cells transported to other sites within the host organism, where they manage to evade phagocytic defenses and establish colonization in new areas. The second dissemination route, independent of phagocytes, occurs via the bloodstream. This strategy benefits from the yeast’s ability to thrive extracellularly and potentially exploit other entry and exit mechanisms. Notably, the mycelial phase of the fungus can traverse epithelial and endothelial barriers in vitro [97].

The swim bladder of *D. rerio* larvae has been ingeniously utilized as a mucosal model to explore *C. albicans* infection dynamics. This approach has shed light on the critical role of neutrophils in restraining the fungus through extracellular traps, leading to hyphal damage. The mycelial phase’s involvement in tissue invasion has been established, whereas macrophages seem to have a comparatively lesser role in controlling mucosal candidiasis [95]. Moreover, this model has unveiled insights into the epidermal growth factor receptor (EGFR) role during *C. albicans* infection. Inhibition of EGFR with the compound AG1478 resulted in larval mortality within 72 h of *C. albicans* inoculation, highlighting the protective function of this receptor in *D. rerio*, with neutrophils playing a significant role in infection control [96].

The yeast and mycelium phases of *C. albicans* both play pivotal roles in host infection and dissemination, with the transparency of *D. rerio* larvae enabling real-time monitoring of the progression of both forms within the host using confocal microscopy [98]. Moreover, as a mucosal model, this model allows a direct comparison of C. albicans and C. parapsilosis infections within the larval swim bladder. The findings indicated that *C. albicans* infections were more severe, resulting in high larval mortality and a robust local proinflammatory response. Both infections triggered macrophage and neutrophil recruitment to the infection site, but *C. albicans* started producing hyphae within three hours post-infection, distinguishing it from *C. parapsilosis*, which remained in the yeast form throughout the observation period, contributing to the more severe outcomes of *C. albicans* infections [99].

The interaction with neutrophils yielded noteworthy differences when comparing infections of D. rerio larvae with *C. auris* and *C. albicans* in the hindbrain. *C. auris* induced limited neutrophil recruitment upon inoculation, while *C. albicans* prompted rapid and substantial neutrophil recruitment. Additionally, *C. auris* seemed capable of evading extracellular neutrophil traps (NETs) and the effects of reactive oxygen species (ROS). Intriguingly, *C. auris* did not seem to activate this vital immune pathway, contributing to its infection severity. The diminished neutrophil recruitment by *C. auris* might be attributed to the mannosylation of its cell wall. In wild-type *C. auris* strains, there was minimal neutrophil recruitment in hindbrain larvae. However, fungal strains with disrupted mannosylation-related genes (pmr1Δ and van1Δ) exhibited significantly enhanced neutrophil recruitment and lower fungal loads in the initial infection phase [97]. These findings underline *D. rerio*’s potency as a host model for exploring different pathogen infections, particularly those of emerging fungal diseases.

*Aspergillus* is another significant fungal genus tested in the *D. rerio* model. This widely prevalent saprophyte can lead to opportunistic lung infections that may disseminate in immunocompromised individuals. *Aspergillus*-associated conditions, such as allergic aspergillosis, often affect individuals with asthma or cystic fibrosis [100,101]. Among the Aspergillus species, *A. fumigatus* is notorious for causing aggressive infections. Different *Aspergillus* species exhibit distinct behaviors and strategies during infection [100].

Using *D. rerio* larvae, researchers investigated the role of LAMMER kinase in aspergillosis, employing a model of T lymphocyte immunosuppression—only the fish’s innate immunity contributed to the results in this approach. The larvae were inoculated with two strains of *A. fumigatus*—wild type and a mutant lacking the LAMMER kinase gene. The absence of the LAMMER kinase gene led to reduced virulence of the fungus, with low mortality observed in all D. rerio groups infected with the *A. fumigatus* LAMMER kinase mutant [101].

Moreover, comparisons were made between *A. fumigatus* and *A. niger* infections in *D. rerio* larvae swim bladder, serving as a mucosal model. The study demonstrated that *A. fumigatus* infections were more severe, leading to high larval mortality and a marked local proinflammatory response. Both infections triggered the recruitment of macrophages and neutrophils to the infection site. However, *A. fumigatus* initiated hyphal growth within three hours post-infection, whereas *A. niger* remained in the yeast form throughout the observation period, contributing to the heightened severity of *A. fumigatus* infections [98,101,102].

Infection with *A. fumigatus* in *D. rerio* embryos and larvae has revealed an intriguing mechanism known as “shuttling”. This phenomenon involves the transfer of phagocytosed microorganisms from neutrophils to macrophages. The process entails contact between cells, leading to membrane fusion (“tethering”), followed by the passage of the phagosome containing the fungus. This interaction occurs in a unidirectional manner, from neutrophils to macrophages. Similar shuttling activity has been observed in infections with the dimorphic fungus *Talaromyces marneffei* using the same *D. rerio* model [103,104].

*D. rerio* has been employed to observe the course of *T. marneffei* infection, with embryos used in a model with only innate immunity. Infections were conducted at 33 °C, where the yeast-like phase of the fungus predominates. Phagocytosis by macrophages was efficient, but neutrophils were found to be the primary cells responsible for clearing the fungus. Despite the role of macrophages as professional phagocytes, they also provide protection, thereby explaining the infection’s spread through the reticuloendothelial system [105].

*Cryptococcus* spp., responsible for lung infections and, in some cases, crossing the blood–brain barrier to cause meningitis, has particularly affected individuals with HIV/AIDS [106,107]. In *D. rerio* larvae, *C. neoformans* infections revealed the yeast’s replication within macrophages, specifically within phagosomes. Expansion of the yeast’s capsule and cell body was also observed during infection, indicating that the fungus can colonize *D. rerio*’s tissues and emulate the disease observed in amphibians [108].

Researchers have demonstrated the pivotal role of macrophages in *C. neoformans* infections. Early in the infection, macrophages phagocytosed yeasts and spores, followed by a period of residence and spore germination. Subsequently, the infection entered a phase of low fungemia, after which yeasts invaded the central nervous system. This stage of low fungemia has also been documented in other models, raising the likelihood of its occurrence in humans. The significance of neutrophils in controlling late-stage infection and the presence of yeasts in endothelial cells were also observed [108].

*D. rerio* has been utilized to study the phenomenon of vomocytosis—a non-lytic expulsion of pathogens by macrophages—during the initial *A. fumigatus* infection. Depletion of macrophages led to increased fungal load, highlighting their essential role in infection control. This unique process further underscores the complexity of host–pathogen interactions [109].

*Mucor circinelloides*, another fungus causing opportunistic infections, belongs to the Mucorales order, responsible for mucormycosis in immunocompromised hosts and frequently affecting patients with decompensated Diabetes mellitus [110,111]. *D. rerio* larvae infected with *M. circinelloides* have demonstrated the formation of granulomas, specialized structures that impede fungal infection progression and reactivate the pro-inflammatory response, supported by ROS, thereby maintaining the infection in a latent state [110].

In adult fish infected with the same fungus, robust recruitment of macrophages and neutrophils from the kidneys—the hematopoietic organ in *D. rerio*—occurred at the infection site. The fungus induced apoptosis in recruited macrophages. Neutrophil mobilization was triggered by both live sporangiospores and those previously inactivated by UV light. Notably, the virulence of the fungus appeared to be linked to sporangiospore size; the R7B strain, producing larger sporangiospores, caused 100% mortality, while the NRRL strain, producing smaller sporangiospores, did not induce mortality [111].

Beyond human pathogens, *D. rerio* has served as a model for studying the amphibian chytridiomycosis caused by the fungus *Batrachochytrium dendrobatidis*. This pathogen has led to the extinction of numerous amphibian species since the 1970s, contributing to the decline of these vital creatures [112]. The development of the *D. rerio* model for chytridiomycosis involved antibiotic-treated (penicillin/streptomycin) fish larvae in an immersion infection system. This approach enabled the observation of infection stages in vivo. Infections were dose-dependent and primarily targeted keratin-rich areas, including the larvae’s caudal fin. Additionally, observations included skin hyperplasia, muscle degeneration, and apoptotic cell accumulation, indicating the fungus’s ability to colonize *D. rerio*’s tissues and simulate the disease seen in amphibians [113].

Leveraging a vertebrate animal’s attributes and the zebrafish’s genetic manipulation capabilities, the *D. rerio* model has proven instrumental in studying infection dynamics. Many techniques can be harnessed to elucidate various facets of pathogen–host interactions within this model.

## 4. The *Caenorhabditis elegans*

*Caenorhabditis elegans* inhabits soil environments (Figure 4) and can also be found in decaying fruits and stems, where it feeds on microorganisms. These diminutive nematodes measure between 825 and 1400 µm in length and 45 to 90 µm in width [114]. *C. elegans* completes its life cycle in a short three days, which is relatively brief. This swift generation time confers an advantage for conducting many assays [115]. Another asset is its transparent body, allowing for tracking its 959 somatic cells. Furthermore, its genome has been entirely sequenced, supported by well-established genetics. This model represents a straightforward manipulation system that does not require ethical licensing, making it an appealing alternative method for various studies [52,116,117,118].

*C. elegans* has emerged as a potent invertebrate model for studying host–parasite interactions and evaluating the virulence of various pathogens, including fungi and bacteria that infect the nematode [118,119]. It is deemed a suitable model for high-throughput screening studies that can be extrapolated to other animal models [120,121,122,123].

Significantly, the *C. elegans* model finds utility in investigating interactions with pathogenic fungi and the emergence of multidrug-resistant fungal strains. The necessity to discover new antifungal agents becomes evident, and this model expedites and streamlines the research process. It functions as a high-performance model capable of assessing numerous compounds to treat fungal infections [123,124].

### 4.1. C. elegans as an Infection Model of Pathogenic Fungi

*C. elegans* served as the pioneering model for investigating host–parasite interactions. In 1960, Brenner reported on this organism’s neural development and gene expression, opening new avenues for research [124,125,126,127]. Over the past two decades, *C. elegans* has emerged as an alternative model for comprehending the biology of various human bacterial pathogens, probing into the nematode’s innate immunity, and exploring its interactions with microorganisms such as *Pseudomonas aeruginosa*, *Staphylococcus aureus*, and *Salmonella enterica* [120,121,128,129].

Regarding fungi, studies have centered on nematode infections induced by dimorphic fungi, yeast-like fungi such as *C. neoformans* and *C. albicans*, and filamentous fungi like *A. fumigatus* and *Drechmeria coniospora*, which commonly inhabit natural settings [123,130,131,132,133,134,135]. Consequently, *C. elegans* has become an extensively explored in vivo model for investigating human infections caused by pathogens [123,124,135,136].

The fungus *A. fumigatus* is the causative agent of aspergillosis, particularly in immunocompromised patients [137,138,139]. In using *C. elegans* to assess *A. fumigatus* infection, researchers identified that the clinical strain Af293 incited infection after 72 h, following a 12 h pre-infection phase [123]. Utilizing the *C. elegans* model, investigators evaluated *A. fumigatus* mutant strains, uncovering pivotal virulence factors like -(1,3)-glucan synthase, melanin pigmentation, transporters, Zn2Cys6-like transcription factors, and mitochondrial thiamine pyrophosphate transporter [123,135]. In assessing antifungal drugs within the *C. elegans* model infected with *A. fumigatus*, conidia germination was inhibited, and this underscored the model’s robust reliability and promising potential for pathogenicity assessment and antifungal agent evaluation [135].

Another study probed the virulence of *C. albicans* when ingested by the nematode. Findings revealed that *C. albicans* hyphae caused nematode demise, with the extent of tissue destruction dependent on fungal virulence. Strains deemed less virulent or with limited hyphae production, such as *Debaryomyces hansenii*, *Candida lusitaniae*, and mutant *C. albicans* strains (efg1/efg1, flo8/flo8, and cph1/cph1 efg1/efg1), led to a slower nematode death, often with scant or no hyphal presence. These observations were consistent with prior murine model results [140]. *Candida* spp., including *C. parapsilosis*, *C. orthopsilosis*, and *C. metapsilosis* curtailed nematode lifespan, highlighting the significance of fungal virulence against the host. These strains prompted intestinal and vulval invasions, culminating in organ protrusion and hyphae formation. Therefore, the *C. elegans* model stands as a viable avenue for studying immediate immune responses and as a model for host–parasite infections [141].

*Drechmeria coniospora*, another fungus studied through this model, initiated its infection process with fungal spore adhesion to the nematode’s head, subsequently leading to extensive hyphal development [142]. Recent research elucidated how *D. coniospora*’s infection model interferes with multiple aspects, inhibiting the activation of the critical STAT transcription factor STA-2, which governs antimicrobial peptide expression. STA-2 levels escalated in the nucleus, inducing nucleolar modifications, and activating *C. elegans*’ surveillance mechanism [143].

Additionally, aside from *Candida* spp. investigations, authors underscored the importance of examining *C. neoformans* strains. *C. neoformans* is recognized for predominantly affecting the central nervous system and causing cryptococcal meningitis but also extends its influence on the lungs and skin [144,145,146]. While *C. elegans* can employ various non-pathogenic yeasts, including *Cryptococcus laurentii* and *Cryptococcus kuetzingii*, as food sources, *C. neoformans* accumulates in the intestine, triggering specific signal transduction pathways (GPA1, PKA1, PKR1, and RAS1) and lactose production (LAC1), ultimately affecting nematode survival. MF1α reporter gene expression in yeast within the nematode gut implies its association with virulence and its direct impact on nematode fate [130,143].

A recent study explored the virulence of the fungal pathogen *C. neoformans* across three distinct hosts: *C. elegans*, *G. mellonella*, and Balb/c mice. Two *C. neoformans* strains, P15 and H99W, were involved, with passages and pre-passages in G. mellonella. Remarkably, all three models exhibited analogous responses. Notably, the P15 lineage exhibited heightened loads in mice and elevated NADH levels, leading to ATP production, metabolic rates, and increased virulence in mice [147].

Concerning dimorphic fungi like *H. capsulatum*, an agent responsible for histoplasmosis, several researchers established an immune response model for *H. capsulatum* infection in *C. elegans* [148,149]. One study unveiled that the fungus caused over 90% nematode mortality within 48–72 h, with the rapid demise suggesting heightened virulence linked to the release of unspecified lethal metabolites [99]. Fungal cells persisted in the intestine, amplifying nematode vulnerability. *C. elegans*’ transparent features facilitated this investigation [127].

While testing *C. elegans* as an infection model for *Paracoccidioides* spp., it became evident that the nematode could not phagocytose the fungi as it did with *H. capsulatum*. Even so, antimicrobial peptides were produced in response to contact with *Paracoccidioides* spp. [150]. These same peptides also increased in the presence of the 14-3-3 protein, indicating their potential use in *Paracoccidioides* spp. as a future vaccine [151].

These studies underscore the significance of alternative models like *C. elegans* and *G. mellonella* in advancing scientific understanding and revealing the innate immune response within parasite–host interactions. While not all fungi are suitable for study within these models, they offer valuable insights into the virulence and pathogenesis of select fungal infections.

### 4.2. C. elegans as a Model for New Antifungal Assessments

The host–parasite interaction observed in *C. elegans* provides a valuable model for screening treatments against microorganisms, assessing new chemicals in vivo, and identifying safer antifungal agents [140]. Utilizing *C. elegans* facilitates the examination of virulence mechanisms expressed by fungi, thereby enabling the redirection of more effective therapeutic strategies [119,130,131,140,152]. Addressing fungal diseases presents a global challenge, notably due to the emergence of drug-resistant strains [153,154]. Reports underscore resistance to azole and polyene antifungals, such as amphotericin B [155,156]. With a restricted arsenal of antifungals, there is a critical need for discovering treatments and, concurrently, more effective models for comprehending the pathogenesis and efficacy of novel compounds [131,152].

A study assessed the antifungal activity of various compounds using a *C. elegans* model infected with *C. albicans* strains constitutively expressing green fluorescent protein (GFP). Post compound administration, nematodes exhibited no fluorescence movement. Of the 1266 tested compounds, 15 were selected, including caffeic acid and phenethyl ester. These compounds effectively inhibited *C. albicans* filamentation, preventing biofilm formation with a minimum inhibitory concentration of 64 μg/mL [157].

Despite *C. albicans* being a primary cause of invasive candidiasis, research has also highlighted *C. glabrata* and other emerging *Candida* species, encompassing clinical isolates of *C. nivariensis* and *C. bracarensis*. Evaluation of antifungal efficacy against *Candida* strains within the *C. elegans* model [136] indicated that amphotericin B and azoles exhibited more significant drug activity against *C. glabrata* and *C. bracarensis* infections, while echinocandins displayed enhanced effectiveness in treating *C. nivariensis* infections. *C. elegans* has emerged as a valuable model for assessing efficacy in treating recurrent infections [158]. Similarly, other investigations involving azoles and polyenes against *A. fumigatus* confirmed the superiority of azole drugs in preventing conidia germination over amphotericin B, further validating the *C. elegans* model [136].

Numerous studies have explored the synergism between compounds potentiated with clinically used antifungals. Investigation into the synergistic impact of tyrosines with amphotericin B and caspofungin against *C. albicans* infection revealed that tyrocidine potentiated caspofungin’s activity, highlighting the potential of tyrocidine in combinatorial therapy [59,153]. Another study examined the synergistic interaction between sulfamethoxazole and voriconazole within the *C. elegans* model. Results demonstrated that the drug combination synergistically increased nematode survival by 70%. Notably, *C. auris* resisted the azole, potentially due to the diminished affinity of the Erg11 target protein. The observed resistance in mutant strains further suggested heightened activity in the microorganism’s efflux pumps [159].

The *C. elegans* model was also utilized to evaluate the probiotic effect against *C. albicans*. *A. baumannii* exhibited the ability to suppress *C. albicans* virulence, resulting in enhanced nematode survival [160]. Additionally, the research explored the combined interaction between *C. albicans* and *Lactobacillus paracasei*, yielding a 29% survival increase and a 27% reduction in nematode cuticle rupture. This combination serves as an alternative control against *C. albicans* [161].

In summary, *C. elegans* represents a high-throughput in vivo screening model for a myriad of antifungal compounds in a liquid medium, substantially reducing the evaluation time compared to conventional models. Accelerating the discovery of novel antifungal drugs is imperative, given the pressing need for new compounds due to escalating fungal resistance and the prevalence of fungal infections. However, while *C. elegans* stands as an attractive model for pathogenesis study and fungal treatment, it is essential to acknowledge its limitations. It cannot entirely replace established mammalian models but is a complementary tool for understanding fungal pathogenesis and discovering innovative antifungal drugs.

## 5. The *Galleria mellonella* Model

This model has emerged as an indispensable tool for studying pathogen–host interactions, generating insights, and devising novel strategies for combating fungal infections. *G. mellonella*, according to some authors, stands as a stellar in vivo model for such investigations, enabling swift, large-scale studies without the need for intricate techniques or specialized equipment for maintenance and treatment [162]. Furthermore, ethical approval is unnecessary. Thriving within temperatures ranging from 25 to 37 °C, the larvae facilitate various studies involving human pathogens. These include survival assays, median lethal dose (LD50) assessment, histopathological examinations, scrutiny of immune system alterations induced by chemical substances and pathogenic infections, evaluation of pathogen virulence, and exploration of behavioral shifts and egg deposition. Already established as a pathogenesis model, *G. mellonella* is also a valuable resource for toxicological assessments in drug discovery. Significantly, the insect’s genome has been sequenced, and its mRNA and transcriptome data are deposited in repositories, paving the way for comprehensive investigations in the pathogen–host interface [163,164,165].

Originating within Apis cerana colonies in eastern Asia, *G. mellonella* has extended its reach worldwide, barring Antarctica. Earning its place as a model in the 1960s [166,167], it witnessed a surge in popularity during the early 2000s concerning studies about bacterial and fungal pathogen–host interactions. The development of diverse methodologies for exploring pathogenesis in *G. mellonella* has contributed to a solid foundational knowledge, thus expanding its utility across various scientific domains [168].

A member of the Lepidoptera order and the Pyralidae family, *G. mellonella*, colloquially known as the “wax worm”, predominantly feeds on beeswax and pollen during its larval stages. The life cycle initiates with butterfly egg deposition, advancing through larval development and culminating in pupation, ultimately leading to adult butterfly emergence, encompassing a 7- to 8-week cycle (Figure 5) [166,169]. This invertebrate model boasts several advantages over mammalian and other in vivo models. Notably, researchers predominantly concentrate on the larval stages, as they facilitate straightforward hemolymph extraction for immune response investigations and melanization reactions against pathogens, inflammation, and stress. Additionally, adult larvae are comparatively more manageable than other life stages. While numerous studies delve into fungal virulence using the injectable route for larval treatment, investigations employing oral and topical routes are equally viable [170,171].

*G. mellonella* is an alternative approach due to its intrinsic immune response, mirroring that of mammals. This response encompasses cellular and humoral components involving antimicrobial peptides, lytic enzymes, peptides, melanin, and hemocyte-mediated processes. The resemblance of wax worm cell defense mechanisms to mammalian neutrophils is noteworthy, exemplified by their capacity to kill and phagocytize pathogens through superoxide production (Figure 6). Intriguingly, the interconnected nature of *G. mellonella*’s cellular and humoral immune systems is evident. Hemocytes assume responsibility for the production of humoral molecules, and their activation by these molecules underscores the model’s significance as an intermediary in vivo system, bridging the gap between in vitro assays and mammalian investigations [168,172].

The wax worm possesses anatomical and physiological barriers fortifying it against the external environment. The cuticle comprises three layers: epicuticle, procuticle, and epidermis. Furthermore, the oral cavity’s limited humidity and nutrient content deter pathogen passage. Even if pathogens breach the intestinal domain, they encounter the deterrent effects of digestive enzymes, pH conditions, and intestinal microbiota. Hemocytes further hinder pathogen spread [168,169,170,171,173].

The gamut of hemocytes underpins the cellular immune response, encompassing phagocytosis, nodulation, and encapsulation of pathogens. In tandem, the humoral response entails the production of antimicrobial peptides, melanization (orchestrated by the phenoloxidase enzyme), hemolymph clotting, primary immunization, and reactive oxygen species [168,170]. These immune warriors occupy the insect’s hemolymph, fat body, and digestive tract, capable of migrating to organs and tissues contingent on immune system activation. The hemocyte brigade encompasses six distinct cell types: prohemocytes, plasmatocytes, oenocytocytes, spherocytes, and granulocytes. Prevalent among them are prohemocytes, assuming the mantle of progenitor cells, while plasmatocytes engage actively in phagocytosis and capsule formation around pathogens. Oenocytoids, non-adhesive entities, bear components of the phenoloxidase cascade implicated in melanization. Spherulocytes undertake the secretion of cuticular components (peptides), although their role remains somewhat enigmatic. Granulocytes prove instrumental in encapsulation and indirectly contribute to phagocytosis [172,174,175,176,177].

Upon encountering pathogenic agents, *G. mellonella* responds with dynamic fluctuations in hemocyte types and quantities contingent on pathogen nature, virulence, and injected cell count. During such scenarios, the cooperative actions of cellular and humoral immune systems converge to curtail microbial proliferation via modulation and phagocytosis, accompanied by escalated granulocyte and hemocyte densities [168,177,178].

Numerous microorganisms have undergone scrutiny within this alternative animal model, encompassing bacteria, fungi, and viruses. Of considerable interest are human fungal pathogens, including *Candida* spp., *Aspergillus* spp., *Mucor* spp., *Cryptococcus* spp., *H. capsulatum*, *Paracoccidioides* spp., and assorted filamentous fungi [179].

### G. mellonella as a Host–Fungi Interaction Model

Many assays can be performed within host–pathogen interactions, leveraging *G. mellonella* larvae. These investigations embrace assessments of larval survival, behavioral alterations, melanization responses, hemocyte quantification, histopathological analyses, and advanced approaches such as proteomics and metabolomics [180]. The amalgamation of these assays furnishes an enhanced grasp of larval immune responses, permitting correlations with mammalian outcomes.

In the context of *Candida* spp., a correlation was drawn between *G. mellonella* and mice infected with thirteen distinct strains of *C. albicans*. Notably, strains that demonstrated avirulence in mice exhibited parallel outcomes within the wax worm model. The investigations unveiled *C. albicans* hyphal formation and yielded congruent results in both models, substantiating the curtailed necessity for mammalian utilization in preliminary research stages. Similarly, other studies compared virulence across *Candida* spp., uncovering the virulence of *C. tropicalis* and *C. albicans* in both *G. mellonella* and murine models. In contrast, other species, including *C. parapsilosis*, *C. krusei*, and *C. glabrata*, displayed minimal lethality within the wax worm context [181,182,183,184,185,186]. Intriguingly, *C. metapsilosis* demonstrated diminished lethality, highlighting its attenuated virulence, whereas *C. orthopsilosis* mirrored comparable mortality rates to those seen in mice [187]. A similar approach was adopted when contrasting *C. auris*—an emergent multidrug-resistant pathogen—to *C. albicans*. Results accentuated the heightened virulence of both *Candida* spp. clinical strains when juxtaposed with reference strains. Additionally, *C. albicans* emerged as more virulent than *C. auris* in *G. mellonella*, evident through mortality and melanization assays [173,188,189]. Augmented virulence in clinical strains could be ascribed to genomic plasticity, leading to elevated virulence factor expression [171,186,190].

Parallels between cellular immune responses (quantification of hemocytes) and humoral immune responses (melanization) and their alignment with pathogenic strains have been explored in depth. Recent studies comparing melanization responses in *G. mellonella* larvae infected with *C. albicans*, *C. auris*, and *C. parapsilosis* strains unveiled no significant disparity in melanization between the three species. However, *C. albicans* exhibited augmented induced melanization, followed by *C. auris* and *C. parapsilosis* [191]. Similar observations emerged while juxtaposing clinical isolates and reference strains of *C. albicans* and *C. auris*, affirming parity in melanization. These studies underscore the imperative of employing multiple assays to evaluate the authentic pathogenicity of medically significant fungi concerning larval immune responses [192,193].

Advanced investigations leveraging proteomics and metabolomics have unearthed novel insights into pathogen–host interplay. Exploring *G. mellonella*’s cellular and humoral immune responses against *C. albicans* uncovered a 60% escalation in larval mortality 24 h post-inoculation, which climbed to 70% after 48 h [177]. The researchers documented a dip in hemocyte density six hours after inoculation, succeeded by a spike at the 24 h mark and a subsequent decline. Notably, larvae treated with β-glucan 24 h post-infection showcased heightened efficacy in *C. albicans* elimination, surpassing the negative control group. Proteomic analysis spotlighted heightened immune-related, muscle, stress-responsive, phenoloxidase-activating proteinase, fungal dissemination, tissue breakdown proteins, and diminished β-1.3-glucan recognition protein precursor and prophenoloxidase subunit 2. These findings suggest that waning hemocyte levels might be tied to nodulation, wherein hemocytes aggregate around pathogenic cells, signifying an imminent cellular and humoral quasi-immune response before larval demise. Comparable outcomes were noted in mammalian models. Other inquiries identified a connection between *C. albicans* genes BCR1, TEC1, FLO8, SUV3, and KEM1 and fungal filamentation, illustrating diminished virulence and pathogenicity of *C. albicans* in *G. mellonella* and mice [163].

The relevance of *G. mellonella* extends to *Aspergillus* spp. infections encompass an array of clinical manifestations, ranging from allergy to aspergilloma and invasive pulmonary aspergillosis. Predominantly attributed to *A. fumigatus*, *A. terreus*, *A. nidulans*, and *A. flavus*, these fungi pose significant risks, particularly to immunocompromised individuals. Studies have demonstrated congruence in the immune responses of *G. mellonella* and murine models during invasive aspergillosis [170]. Investigations underscored heightened hemocyte density within *G. mellonella* two hours post-*A. fumigatus* infection, persisting at 6, 8, and 10 h. In mice, *A. fumigatus* infection yields analogous outcomes of augmented neutrophils and macrophages. CryoViz imaging and fluorescence microscopy unveiled *A. fumigatus* spread within *G. mellonella*’s distal regions, forming melanized nodules around fungal cells. Early responses encompass encapsulation, nodulation, and melanization. These findings suggest a nexus between melanized nodules in wax worms and granulomatous structures observed in mice during invasive aspergillosis. Proteomic analyses identified heightened antimicrobial peptides and proteins tied to the prophenoloxidase cascade at the six-hour mark post-inoculation. Moreover, a 24 h milestone marked an upsurge in invasion, opsonization, and recognition-related proteins, coupled with diminished fungal proteinase activity. The study parallels *G. mellonella* and mammalian outcomes during invasive aspergillosis, contributing to model validation [194,195].

Mucormycetes, counting *Mucor* spp., *Rhizopus* spp., *Lichtheimia* spp., and *Rhizomucor* spp., are of notable clinical significance. These fungi frequently underlie cutaneous and deep-tissue infections, particularly within immunocompromised individuals. Of late, *Mucor* spp. have garnered attention amid the COVID-19 pandemic. Noteworthy challenges encompass variable drug susceptibility, complex diagnosis, and prognosis due to limited insight into their pathogenic mechanisms [196,197]. Utilizing *G. mellonella*, researchers assessed virulence data and standardized the impact of medium composition on strain virulence concerning Mucormycetes infections [198]. Through survival analysis, histological evaluation, quantification of hemocyte density, drug susceptibility assessments, and standardized infection methodologies, the researchers demonstrated the viability of *G. mellonella* as an additional in vivo model for studying mucormycetes virulence factors and drug pharmacodynamics.

Similarly, *Paracoccidioides* spp. and *H. capsulatum*, prominent dimorphic fungi and causal agents of paracoccidioidomycosis (PCM) and histoplasmosis, respectively, have attracted extensive attention. PCM chiefly impacts tropical regions, particularly Latin America’s Brazil, Argentina, Colombia, and Venezuela. On the other hand, histoplasmosis has global prevalence, extending across the United States, Africa, and numerous Latin American locales such as Brazil, Colombia, Argentina, and Venezuela. Both fungi can induce systemic mycoses, culminating in dire consequences if not swiftly addressed [196,197]. Between 1996 and 2006, PCM stood as Brazil’s leading cause of death, closely followed by other mycoses like cryptococcosis, candidiasis, and histoplasmosis [199]. Similarly, histoplasmosis is the fourth most common systemic mycosis, impacting over 500,000 individuals annually [1].

Exemplary studies have harnessed *G. mellonella*’s infection potential to evaluate the efficacy and toxicity of antifungal drugs against *P. brasiliensis* and *P. lutzii*. Utilizing survival analyses, fungal burden assessments, histological examinations, and hemocyte density quantification, these investigations illuminate *G. mellonella*’s potential in drug screening and discovery [200,201,202]. Moreover, the exploration of *H. capsulatum* delved into the impact of heat shock protein 60 kDa (Hsp60)—a widely expressed fungal protein during biofilm formation—on *G. mellonella*. Treating *H. capsulatum* biofilm-derived cells with monoclonal antibody anti-Hsp60 before inoculation bolstered larval survival by 60%. This insight into Hsp60’s role underscores its significance in the host–pathogen interplay, offering a potential target for novel drug development [203].

Numerous clinically relevant fungal pathogens have encountered scrutiny within the *G. mellonella* model, unraveling fresh perspectives on drug discovery and host–pathogen interactions. The capacity to extrapolate data to mammalian systems further elevates the model’s worth, offering comprehensive insights by amalgamating various assays.

## 6. The *Tenebrio molitor* Model

*Tenebrio molitor*, commonly known as the mealworm, falls under the order Coleoptera and the family Tenebrionidae. It is a widespread invertebrate found worldwide. The life cycle of *T. molitor* consists of four stages: the embryonic (eggs), larval, pupal, and adult stages (Figure 7). Notably, the larval stage can encompass up to 20 instars after molting [204,205]. Much like *G. mellonella*, *T. molitor* can be cultivated at 25 and 37 °C, rendering it a versatile model for studying microbes at environmental and mammalian temperatures. Additionally, the publication of *T. molitor*’s transcriptome has opened doors for molecular biology studies [206]. *T. molitor*’s innate immune response is composed of both cellular and humoral components. The cellular innate immune system employs Toll-like receptors to recognize pathogens, enabling phagocytosis and the development of autophagy, encapsulation, and nodulation processes. Antimicrobial peptides (AMPs), including tenecins and melanin, are produced in response to infection [1,2,3,4,207]. Interestingly, the innate immune response in *T. molitor* can endure even without the immunogenic agent, resembling the persistence observed in vertebrate adaptive immune responses [208]. *T. molitor* synthesizes several AMPs to defend against microbial pathogens, including fungi [209].

*T. molitor* has proven to be a valuable model for assessing fungal virulence and host–pathogen interactions. Comparative studies involving clinical isolates of *A. fumigatus* demonstrated decreased survival rates in *T. molitor* larvae with increasing fungal inoculum concentrations. Lower inoculum concentrations corresponded to reduced fungal burden, suggesting fungus shedding despite heightened larval mortality [210]. Similarly, the virulence of different species of *Malassezia* was evaluated in the *T. molitor* model. Larvae inoculated with *M. furfur* grown in a lipid-supplemented medium exhibited enhanced survival [211,212].

Research into *Sporothrix* species demonstrated that *S. schenckii* yeast-like cells were the most effective fungal morphotype in killing *T. molitor* larvae. This effect was dose-dependent, and low-virulence strains exhibited limited cytotoxicity, phenoloxidase activity, and hemocyte counting. Using a recombinant protein (Gp70) from *S. schenckii* stimulated a protective effect against lethal doses of various *Sporothrix* spp. [213]. *T. molitor* has also been employed in studying the causative agent of chromoblastomycosis. Fornari et al. investigated strains of *Fonsecaea* spp., confirming the pathogenic adaptation of specific morphotypes in animal tissues [213].

Further studies with *T. molitor* explored two other significant human pathogenic fungi: *C. albicans* and *Cryptococcus* spp. Increased fungal inoculum concentrations led to higher larval mortality, and observations highlighted the growth of *C. albicans* mycelia and *C. neoformans* yeast and capsule visualization using staining techniques [214,215]. The antifungal activity of a hexane extract from *Spondias tuberosa* leaves demonstrated a survival benefit in *C. gattii*-infected *T. molitor* larvae, indicating potential anticryptococcal effects [214,216].

The *T. molitor* model has also been instrumental in investigating host–pathogen interactions involving dimorphic fungi like *Paracoccidioides* spp. Research focusing on iron acquisition by *P. brasiliensis* revealed the importance of this process in the fungus’s virulence, as *T. molitor* larvae infected with a siderophore biosynthesis enzyme mutant exhibited higher survival rates [215].

The functional roles of immune proteins in *T. molitor*’s defense mechanisms have been examined. Silencing the TmSR-C gene in hemocytes using RNAi highlighted the protein’s significance for the insect’s survival during *C. albicans* infection, emphasizing its phagocytic role in the immune system [216]. Similarly, the RNAi technique was utilized to explore the protective role of Tm14-3-3ζ against *C. albicans* infection [217].

The studies mentioned underscore the potential of *T. molitor* larvae as a model to investigate various aspects of fungal infections, such as host–fungal interactions, immune responses, fungal virulence, mycotoxin effects, and the efficacy of antifungal compounds. Despite the progress made, there remains a need for further research to address existing gaps in our understanding. *T. molitor* has potential as an alternative host model for advancing our knowledge of fungal infections and immunity.

## 7. Conclusions

We present a comprehensive exposition of diverse applications employing alternative methodologies to mammalian subjects in scientific inquiry concerning fungus–host interactions. These alternative models offer considerable contributions towards reducing reliance on mammalian organisms, concurrently fostering substitution and refinement of established methodologies.

Progressions in technology, particularly within the domain of three-dimensional cultures, substantively contribute to the generation of models that progressively emulate human physiological conditions. Nonetheless, there is a need for further advancements, particularly in the domain of intricate models encompassing a symbiotic microbiome. These models hold the potential for evaluating the intricate interplay of pharmacokinetics and pharmacodynamics pertinent to novel therapeutic agents.

In juxtaposition to conventional in vivo models, these alternative paradigms present ethical merits and facilitate the swift acquisition of extensive datasets. They retain the organizational intricacies inherent to living organisms and evince outcomes akin to those observed in mammalian subjects, particularly in the preliminary stages of novel drug investigations. Consequently, these models expedite the temporal trajectory of drug development, encompassing pharmaceutical agents and the broader domain of fungus–host interactions. Their utility encompasses validating therapeutic targets through mutational studies, appraisal of clinical strain virulence, identification of diagnostic markers, and assessing therapeutic efficacy and safety profiles.

Despite these strides, there remains substantial ground to be covered in pursuing models that more faithfully emulate the human milieu. These future models should be characterized by ethical viability, robust reliability, and economic feasibility.

## Figures and Tables

**Figure 1 jof-09-00943-f001:**
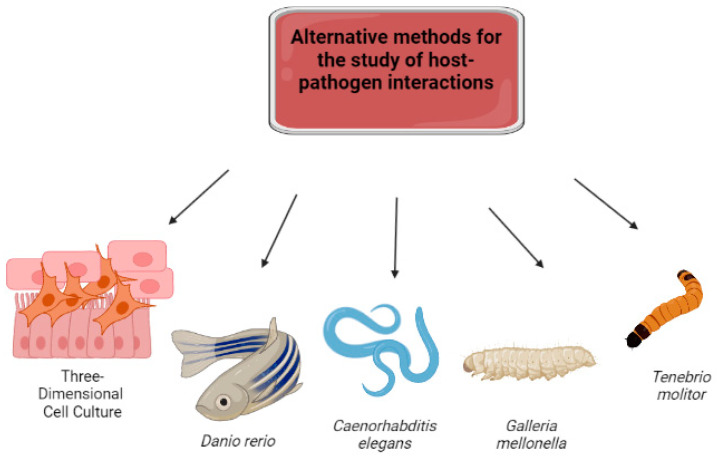
Main alternative non-mammalian animal and cellular methodologies to replace vertebrates for studying pathogen–host interactions. Resource: own resource.

**Figure 2 jof-09-00943-f002:**
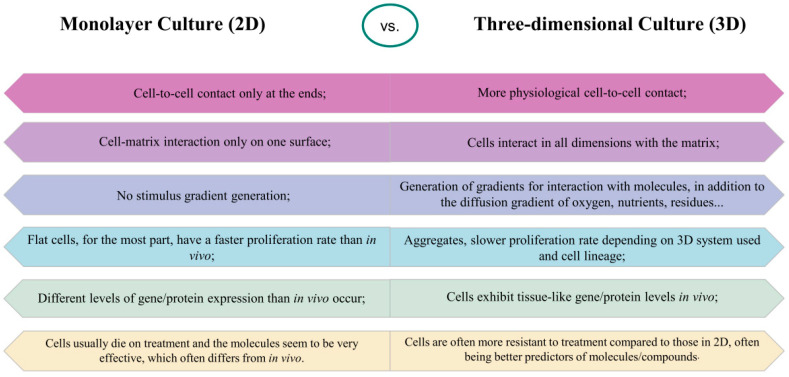
Main differences between monolayer (2D) cultures and three-dimensional (3D) models. Adapted from: [37]. Resource: own resource.

**Figure 3 jof-09-00943-f003:**
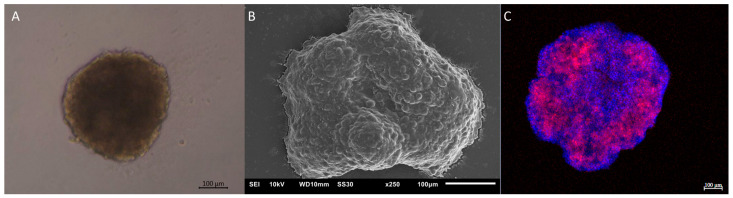
Representative images of the spheroids formed by the lung cell line A549 and MRC-5. (**A**)—A549 cell line observed in bright field microscopy with 10× magnification; (**B**)—spheroid of the A549 cell line in the ×250 objective obtained by scanning electron microscopy and (**C**)—fluorescence of the three-dimensional culture of the MRC-5 cell line—the blue stain shows the nucleus of the cells and in red the cytoplasm of the cells. It was adapted from [26] and its authorship. Resource: own resource.

**Figure 4 jof-09-00943-f004:**
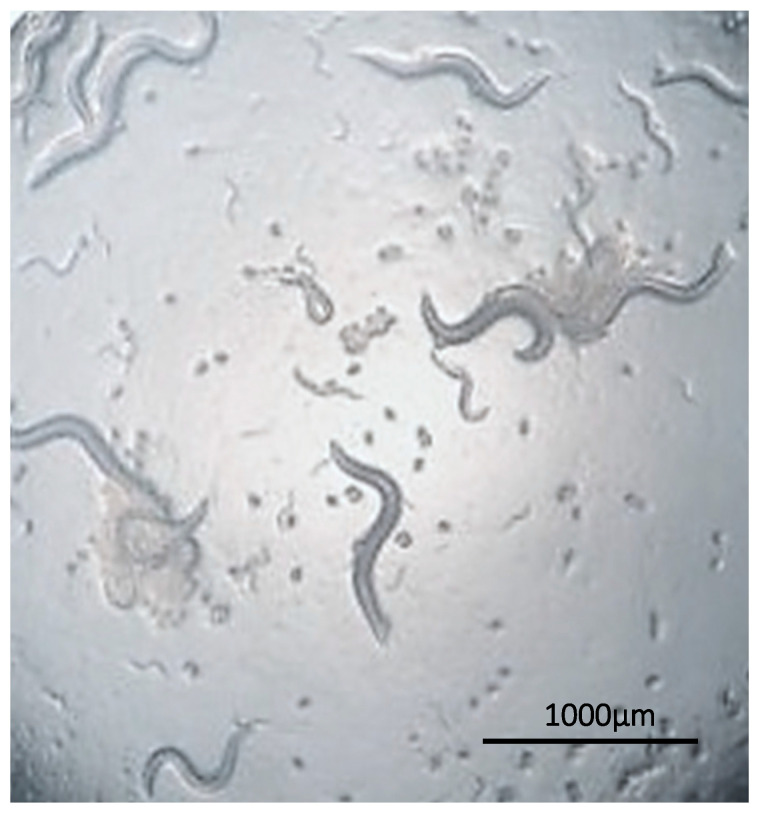
Representative image of the *C. elegans* larvae in different L1, L2, L3, and L4 larvae stages and eggs. Resource: own resource.

**Figure 5 jof-09-00943-f005:**
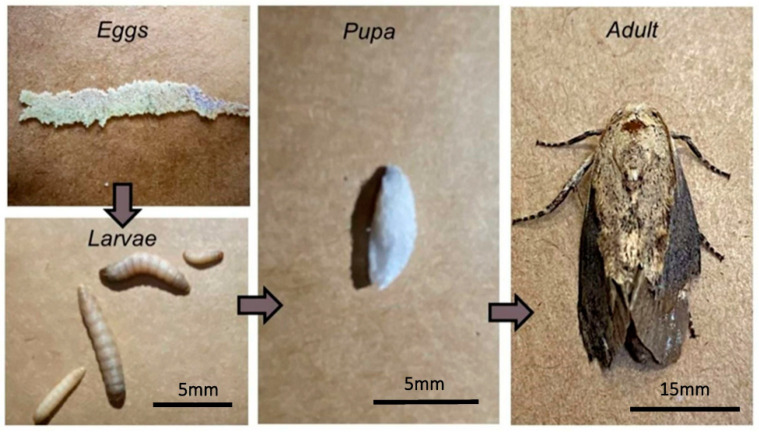
The life cycle of *G. mellonella*. The life cycle starts with adults depositing eggs. After hatching, the larvae enter the larval phase, which lasts approximately three weeks. They then form pupae, which eventually develop into adult butterflies. Resource: own resource.

**Figure 6 jof-09-00943-f006:**
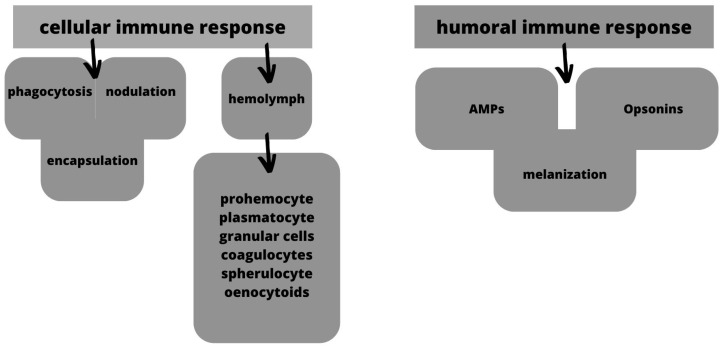
*Galleria mellonella* immune system. *Galleria*’s adaptative immune response is divided into cellular and humoral responses, mediated by hemocytes. Cellular immune response refers to hemocytes phagocytosis, encapsulation, and nodulation of pathogens. In contrast, the humoral immune response refers to hemocytes’ production of opsonins, antimicrobial peptides (AMPs), and melanization. Resource: own resource.

**Figure 7 jof-09-00943-f007:**
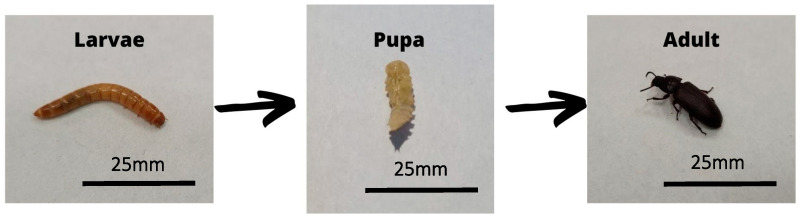
Different stages of the worm *Tenebrio molitor* are represented by larvae, pupa, and adults. Resource: own resource.

## Data Availability

Not applicable.

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
