# Peer review of "Alternative Non-Mammalian Animal and Cellular Methods for the Study of Host–Fungal Interactions"

_jof, 2023, doi:10.3390/jof9090943_

Round 1
Reviewer 1 Report
The authors have written a very comprehensive review of the use of non-vertebrate models to explore fungal pathogenesis. They use three “R’s” as aspirational goals for this type of proposal, and this platform is a good way to structure the review.
They provide a detailed summary of each model system, transitioning to how they have been used in studies on fungal pathogenesis. They are both comprehensive and balanced in their review of current literature. They also offer visions for future directions for each system.
Minor points:
Line 74. I do not understand “a usual behavior repertoire”
Line 111. Are 3-D cultures truly “widely used” at this point. While promising for all the reasons discussed, I see them as an emerging and underutilized resource that takes much more skill to create.
Line 205. Is “intense” the most appropriate word? Not sure the meaning implied.
Line 312. What is meant by zebrafish being “resistant fish”?
Line 402. Italicize the mutant gene names.
Line 461. Change “evolution” to “dissemination” or similar.
Line 541. Italicize C. elegans
Line 657. I agree that, despite its promises, there are significant limitations to C elegans for drug screening. This needs more discussion.
Line 794. Italicize A fumigatus
Line 829. Italicize H capsulatum
In the first figure, they do not include 2D cell culture or organoids, although they are discussed in the text distinct from 3D cell culture?
Fig 4 legend. Italicize C elegans.
Author Response
Minor points:
Line 74. I do not understand “a usual behavior repertoire”
Answer: Modified on the manuscript, the sentence was rewritten for better understanding
Line 111. Are 3-D cultures truly “widely used” at this point. While promising for all the reasons discussed, I see them as an emerging and underutilized resource that takes much more skill to create.
Answer: Despite understanding the placement, among the three-dimensional cultures there are methodologies that are relatively easy to perform, such as the development of spheroids for example, to perform the technique the researcher would mostly need only agarose to coat the plate and its cell of interest. With that in mind, methodologies such as this one was included in the review, covering both less sophisticated techniques and more complex ones.
Line 205. Is “intense” the most appropriate word? Not sure the meaning implied.
Answer: Modified on the manuscript, the sentence was rewritten for better understanding
Line 312. What is meant by zebrafish being “resistant fish”?
Answer: Modified on the manuscript, the sentence was rewritten for better understanding
Line 402. Italicize the mutant gene names.
Answer: modified as requested
Line 461. Change “evolution” to “dissemination” or similar.
Answer: modified as requested
Line 541. Italicize C. elegans
Answer: modified as requested
Line 657. I agree that, despite its promises, there are significant limitations to C elegans for drug screening. This needs more discussion.
Answer: two paragraphs were added, clarifying a little more about this
Line 794. Italicize A fumigatus
Answer: modified as requested
Line 829. Italicize H capsulatum
Answer: modified as requested
In the first figure, they do not include 2D cell culture or organoids, although they are discussed in the text distinct from 3D cell culture?
Answer: Thanks for the comment, the image has been modified according to what the text presents, new figures have been added
Fig 4 legend. Italicize C elegans.
Answer: Answer: modified as requested
Reviewer 2 Report
A deeper review and comparisons should be made to the concerned topic. The authors should consider the followings:
More workflow mechanistic demonstration of the host-fungal interaction of each animal models and of cellular model should be employed.
The authors should review and list the limitations of using the concerned non-mammalian animal models and of cellular model, vs mammalian models and vs human clinical trials.
Instead of generally summarizing the 2D vs 3D models, the authors should focus on the use of host-fungal interaction between 2D and host-fungal interaction between 3D.
In Figure 4 and 5, scale bars should be added to the figures. In Figure 4, the authors should use arrow points to locate the stages of the larvae, and to use a more magnified images to show the morphologies.
In Figure 7, scale bars should be added to the figures.
By ADME and PK/PD perspectives, the authors may compare the said models to the conventional animal models.
The authors may discuss which areas of host-fungal interaction, could be of advantages using the said non-mammalian animal models and of cellular model, vs mammalian models.
The authors may tabulate to compare the systems between non-mammalian animal models and of cellular model, vs mammalian models, across host-fungal interaction.
Quality of English Language should be assessed by English Professional.
Author Response
1. More workflow mechanistic demonstration of the host-fungal interaction of each animal models and of cellular model should be employed.
Answer: Without modification after reviwed.
2. The authors should review and list the limitations of using the concerned non-mammalian animal models and of cellular model, vs mammalian models and vs human clinical trials.
Answer: Changed as request.
3. Instead of generally summarizing the 2D vs 3D models, the authors should focus on the use of host-fungal interaction between 2D and host-fungal interaction between 3D.
Answer: The differences between the models are the main point that makes the host-pathogen interaction in 3D models more reliable, as justified in the text.
4. In Figure 4 and 5, scale bars should be added to the figures. In Figure 4, the authors should use arrow points to locate the stages of the larvae, and to use a more magnified images to show the morphologies.
Answer: Changed as requested.
5. In Figure 7, scale bars should be added to the figures.
Answer: Changed as requested.
6. By ADME and PK/PD perspectives, the authors may compare the said models to the conventional animal models.
Answer: Alternative animal models have not been compared to mammalians in terms of PK/PD perspectives, and studies of this nature are scarce. Even mammalians can predict a limited reliability and predictive of PK/PD in humans (Akhtar A. The flaws and human harms of animal experimentation. Camb Q Healthc Ethics. 2015 Oct;24(4):407-19. doi: 10.1017/S0963180115000079. PMID: 26364776; PMCID: PMC4594046). When alternative animal models are used in initial drug development and host-pathogen interaction studies the consense is that they contribute to replacing, reducing, and refining the researching decreasing the number of mammals used to its purpose (Boyd WA, Smith MV, Co CA, Pirone JR, Rice JR, Shockley KR, Freedman JH. Developmental Effects of the ToxCast™ Phase I and Phase II Chemicals in Caenorhabditis elegans and Corresponding Responses in Zebrafish, Rats, and Rabbits. Environ Health Perspect. 2016 May;124(5):586-93. doi: 10.1289/ehp.1409645. Epub 2015 Oct 23. PMID: 26496690; PMCID: PMC4858399).
7. The authors may discuss which areas of host-fungal interaction, could be of advantages using the said non-mammalian animal models and of cellular model, vs mammalian models.
Answer: The use of alternative methods to study host-pathogen interaction was discussed in the text, they are related to the 3Rs principles in research, contributing to reducing and replacing mammalians and as a consequence, they are more ethical but also give reliable answers in initial trials in pre-clinal and clinical (phase I) research. All the topics are in the manuscript and a conclusion was inserted into the manuscript.
8. The authors may tabulate to compare the systems between non-mammalian animal models and of cellular model, vs mammalian models, across host-fungal interaction.
Answer: Without modification after reviwed.